# Graz Lagrangian Model (GRAL) for Pollutants Tracking and Estimating Sources Partial Contributions to Atmospheric Pollution in Highly Urbanized Areas

**Aleksey A. Romanov** [1,2,*] **, Boris A. Gusev** [1,2] **, Egor V. Leonenko** [1] **,**
**Anastasia N. Tamarovskaya** [1,2] **, Alexander S. Vasiliev** [1] **, Nikolai E. Zaytcev** [1,2] **and**
**Ilia K. Philippov** [1,2]

1    A[2] Research and Development Lab, 660011 Krasnoyarsk, Russia; BGusev@a2rd.com (B.A.G.);
     ELeonenko@a2rd.com (E.V.L.); ATamarovskaya@a2rd.com (A.N.T.); AVasilev@a2rd.com (A.S.V.);
     NZaycev@a2rd.com (N.E.Z.); IPhilippov@a2rd.com (I.K.P.)
2    GIS Department, Siberian Federal University, 660041 Krasnoyarsk, Russia
*    Correspondence: ARomanov@a2rd.com; Tel.: +7-902-910-0099

**Abstract:** Computational modeling allows studying the air quality problems in depth and provides the best solution reducing the population risks. This research demonstrates the Graz Lagrangian model effectiveness for assessing emission sources contributions to the air pollution: particles tracking and accumulation estimate. The article describes model setting up parameters and datasets preparation for the analysis. The experiment simulated the dispersion from the main groups of emission sources for real weather conditions during 96 h of December 2018, when significant excess of $NO_2$, CO, $SO_2$, PM10, and benzo(a)pyrene concentrations were observed in the Krasnoyarsk surface atmospheric layer. The computational domain was a parallelepiped of $40 \times 30 \times 2.5$ km, which was located deep inside the Eurasian continent on a heterogeneous landscape exaggerated by high-rise buildings, with various pollutions sources and the ice-free Yenisei River. The results demonstrated an excellent applicability of the Lagrange model for hourly tracking of particle trajectories, taking into account the urban landscape. For values < 1 MPC (maximum permissible concentration) of peak pollutants concentrations, the coincidences were 93 cases, and for values < 0.1 shares of MPC, there were 36 cases out of the total number of 97. The same was found for the average daily concentration for values < 1 MPC—31, and for values < 0.1 MPC—5 matches out of 44. Wind speeds COR—65.3%, wind directions COR—68.6%. The Graz Lagrangian model showed the ability to simulate air quality problems in the Krasnoyarsk greater area conditions.

**Keywords:** air quality; air dispersion modeling; GRAL; Lagrangian model; air pollutants tracking; computational fluid dynamics; model performance evaluation

## 1. Introduction

Filthy air is a severe problem and a challenge for industrialized megacities worldwide [1]. Correctly responding to such difficulties requires analytical systems that accurately determine the sources significantly impact on the pollution within the lower atmospheric layers. They are needed first for making proper urban planning and other decisions by local authorities and secondly, for the introduction or intensification of effective environmental protection measures at the industrial enterprises [2]. Any network of sensors by itself cannot determine the source of pollution. Therefore, there is no alternative to computational methods, and for this reason, they are broadly used in

practice [3,4]. The accuracy and the speed of assessment (obtaining of the results) are essential for computational models. The Box model, Gaussian, Lagrangian, the PUFF, Eulerian and Computational Fluid Dynamics (CFD) are most commonly used [5–8]. Recommended approaches for different scales and applications of atmospheric dispersion modeling are listed in Table 1.

**Table 1.** Recommended approaches for different scales and applications of atmospheric dispersion modeling [1].

| Application | <1 km | 1–10 km | 10–100 km | 100–1000 km |
|---|---|---|---|---|
| Online risk management (short runtime is important) | - | Gaussian | Puff | Eulerian |
| Complex landscape | CFD | Lagrangian | Lagrangian | Eulerian |
| Reactive materials | CFD | Eulerian | Eulerian | Eulerian |
| Source–receptor sensitivity | CFD | Lagrangian | Lagrangian | Lagrangian |
| Long-term average loads | - | Gaussian | Gaussian | Eulerian |
| Free atmosphere dispersion (volcanoes) | - | Lagrangian | Lagrangian | Lagrangian |
| Convective boundary layer | CFD | Lagrangian | Eulerian | Eulerian |
| Stable boundary layer | CFD | Lagrangian | Eulerian | Eulerian |
| Urban areas, street canyon | CFD | CFD | Eulerian | Eulerian |

[1] Cited from [5].

There are several techniques that aimed to solve the backward transfer problem from the measuring point [9]; these methods utilize an atmospheric dispersion model, which is used to determine the location of the pollution source with known wind parameters. Backtrack methods obtained improved results when they were combined with forward dispersion methods. However, due to the high complexity and uncertainty, they are still inferior to the direct methods presented above. For example, in study [10], the Stochastic Lagrangian particles distribution model showed the correlation coefficient (COR) 0.89, while for the backward calculation, COR = 0.62.

Ground-level air pollution model (GLAPM) is applied in the Russian Federation for estimating the emission dispersion (formalized as Gaussian-based method; described in the law-status document "Ministry of Natural Resources Order #273 of 6 June, 2017") [11]. Approaches based on this model have serious shortcomings, since it does not consider the following factors effecting on the spread of pollution:

- Complex landscapes (geomorphology) and landcover types. They define the local heat balance and surface air flows because of different albedo, vegetation, unevenness, and thermal conductivity [12,13].
- Buildings (an urban landscapes). High-rise buildings exaggerate the urban landscape and influence on the wind flows [14–16].

The GLAPM model uses many parameters in constants, which dramatically simplifies the nature of wind impacts on pollutant dispersion within cities. Atmospheric inversion and local surface air flows differ significantly from the general meteorological dataset represented by the GLAPM. For instance, wind speeds of less than six m/s in this model are set to equal six m/s. This simplification makes this model unacceptable for many cities. The approaches based on this model allow determining the average annual pollutants concentration at an acceptable level for simple scenarios. Nevertheless, for specific regional conditions, they are inadequate, especially in short-term periods. As a result, management decisions are often based on gross emissions estimates that do not reflect reality, and ultimately, these measures are ineffective.

Air quality problem is very relevant for Krasnoyarsk city (Russian Federation)—it is an industrialized metropolis with many emission sources, surrounded by mountains, located deep inside Eurasia. The absence of wind covers the city by a smog-shroud just in several hours (Figure 1). Regulatory authorities and ecologists often record the excess of pollutants indicators, such as

Benzo(a)pyrene, NO$_2$, CO, SO$_2$, and PM10 [17,18]; the city is often included in the top global air quality index anti-rating [19], and this problem seriously escalated in the society (the activists widely use "the black sky" ideologeme attracting attention to the problem) [20]. The major management decisions are made based on the air quality monitoring network and a regulatory document on the gross emissions limits [21].

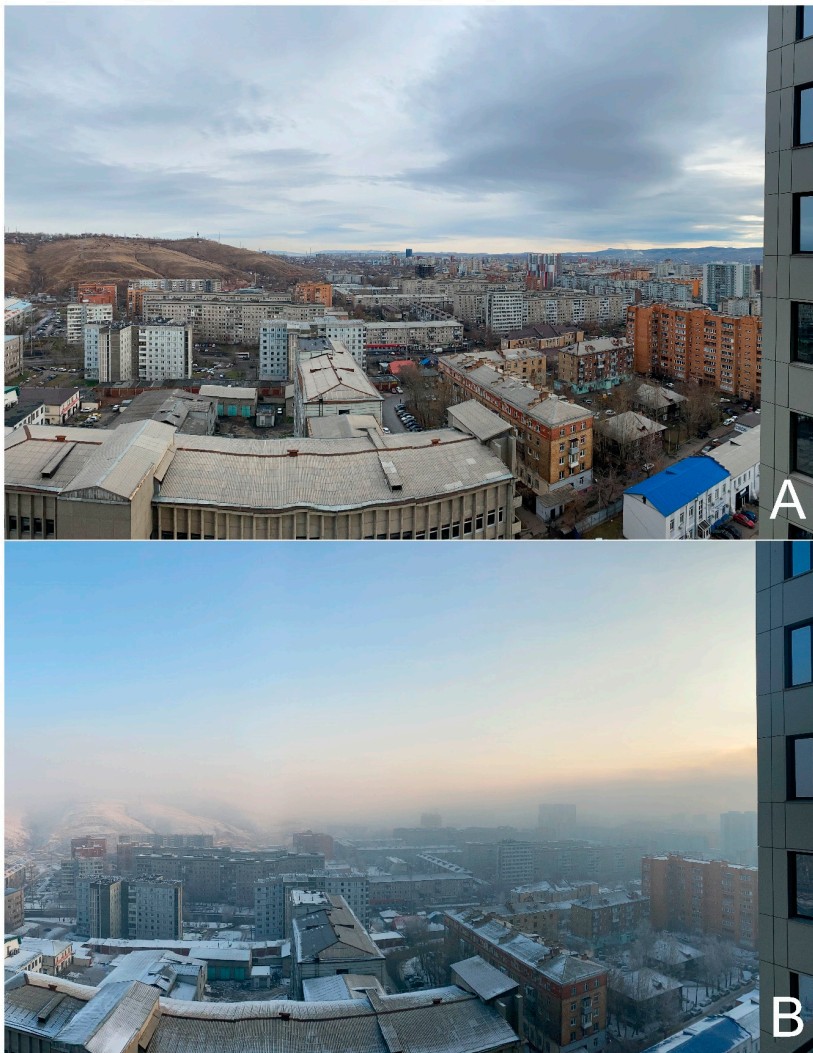

**Figure 1.** Krasnoyarsk city view from a high elevation point: (**A**)—2 November 2020 (wind direction—SW, wind speed 2 m/s); (**B**)—28 November 2020 (calm, air temperature −14 Celsius, humidity—85%, atmospheric pressure 1040 mb, AQI from 175–638). Credit: A Romanov.

Several studies were devoted to the Krasnoyarsk's air problems [22,23], and some of them pointed out the disadvantages of the applied methodological approaches based on the GLAPM [24]. The problem of the Yenisei River, which is ice-free during the winter seasons and impacts on the wind flows, was also highlighted using the CFD modeling in [25].

The objectives of the current research were to assess the pollutants concentrations and determine the contributions of the main groups of sources to Krasnoyarsk's atmosphere, concerning the regional specifics. Based on the research results in [26–28], which demonstrates the advantages of Lagrangian models for complex landscapes and communities, the Graz Mesoscale Model (GRAMM)/Graz Lagrangian model (GRAL) methodology [29] was used to simulate the pollutants dispersion from the sources (industry, heat-generating facilities, transport, and private households). A computational experiment was carried out based on the real meteorological parameters for the particular period

(the eve of the 2019 Winter Universiade [30]). Part of the research is devoted to the Krasnoyarsk greater area specifics (domain size is 40 × 30 km); a detailed input data description and the model settings are presented; the obtained results are compared with the official statistics data.

## 2. Experiments

Krasnoyarsk city is a complex research object that requires several studies of its specific to prepare the initial data for modeling.

### 2.1. Krasnoyarsk Greater Area Specifics

Krasnoyarsk is a developed industrial center with over one million people. The problem of the city's air quality deteriorates significantly during the periods of adverse weather conditions (AWC), which are characterized by weak regional wind flows, high atmospheric pressure, and low overcast. This issue is due to a combination of natural and anthropogenic factors.

Natural ones include the following:

- A heterogeneous landscape (eight high-rise terraces; elevation from 117 to 708 m.a.s.l.); the city is located in a hollow, surrounded by the hills—spurs of the Sayan mountains (Figure 2);
- Continental climate contrasts with the ice-free Yenisei River—the effect of evaporation seriously impacts on the air flows and complicates modeling due to the significant temperature difference of water and the environment (even at −40 degrees Celsius, the river bed is not covered with ice within the boundaries of the agglomeration because of the hydroelectric dam 30 km upstream; the area of non-freezing surface is 29 km$^2$);
- The Krasnoyarsk greater area is located on the periphery of the Siberian anticyclone [31], which is probably why the average daily wind speeds have significantly decreased during the last four decades, while the number of calms has increased from 2002 to 2019 (Figures 3 and 4).

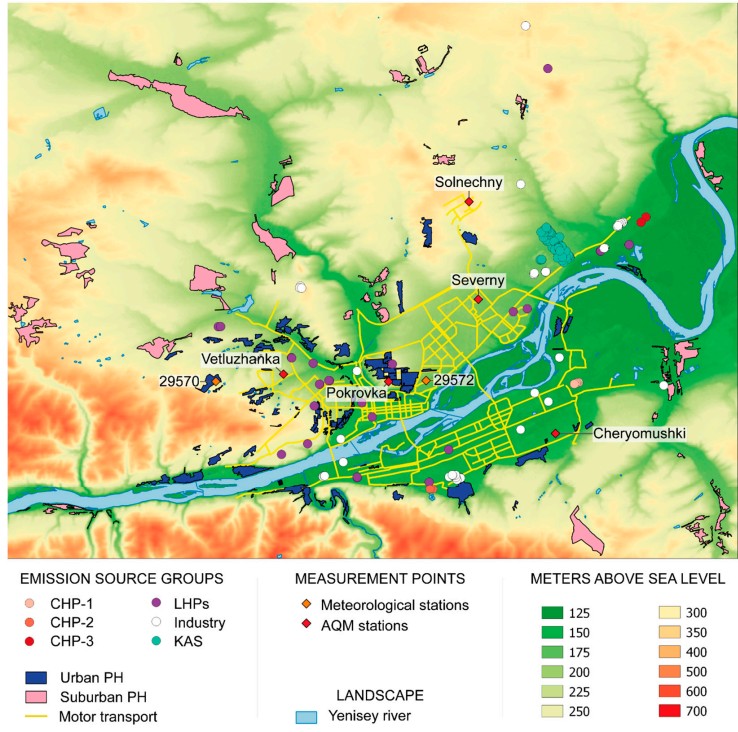

**Figure 2.** A heterogeneous landscape of the Krasnoyarsk greater area. Spatial distribution of the main emission sources, meteorological stations, and observation points (air quality measuring stations, AQM).

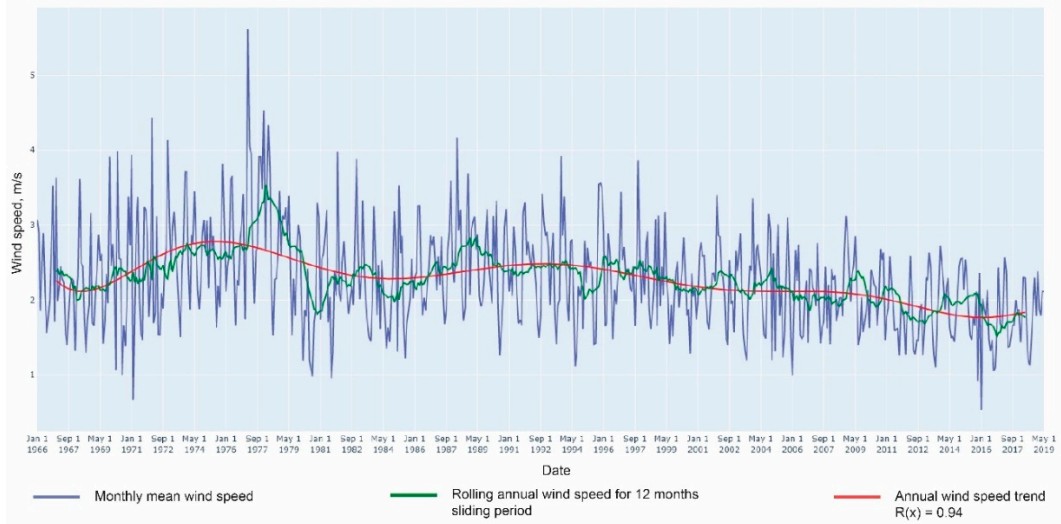

**Figure 3.** The average monthly wind speeds for the period from 1966 to 2019.

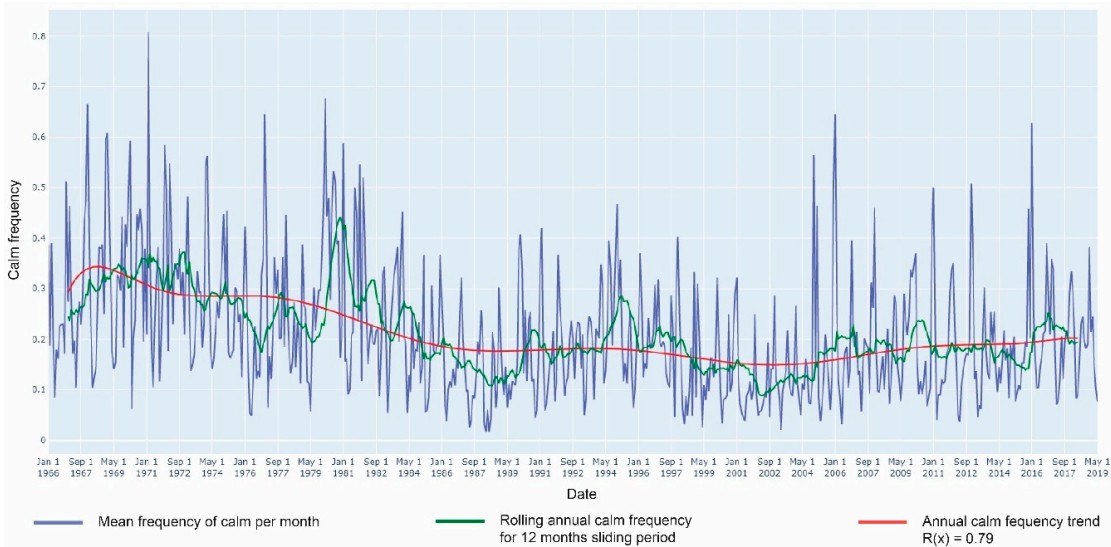

**Figure 4.** Calm frequency for the period from 1966 to 2019.

The graphs of the average daily wind speeds and calms frequencies built based on the data of meteorological station #29570 from All-Russia Research Institute of Hydrometeorological Information—World Data Centre (RIHMI-WDC) [32,33]. Trends were obtained based on specially developed Equations (1) and (2):

$$\overline{v}_j = \frac{1}{q_j r_j} \sum_{s=0}^{q_j} \sum_{t=0}^{r_j} v_{m_{st}}, \{v_m : v_m \in Z \land v_m > 0\} \tag{1}$$

where $\overline{v}_j$—average wind speeds per month, m/s; $q_j$—number of days relative to month; $r_j$—number of measurements relative to the day; $s$—day index relative to month; $t$—measurement index relative to the day; $v_{m_{st}}$—wind speed relative to measurement (every three hours) and day, m/s; $v_m$—wind speed measurements set, m/s.

$$f_j = \frac{k_j}{n_j} = \frac{|\{v_m : v_m \in \mathbb{Z} \land v_m = 0\}|}{|\{v_m : v_m \in \mathbb{Z} \land v_m \geq 0\}|}, \tag{2}$$

where $f_j$—calm frequency relative to month and year; $k_j$—number of calms among all measurements per month; $n_j$—number of measurements per month; $v_m$—wind speed measurements set, m/s; $j$—month number relative to year.

The anthropogenic impact is due to the following aspects:

- Emissions dispersion deterioration in the surface layer as high-rise buildings exaggerate the landform;
- The private sector households heated individually without a centralized system (by lignite or firewood).
- Gross emissions are increasing by heating and energy supply because of the civil construction growth.
- The increasing number of vehicles contributes to the rise in gross emissions, as well as the loads on roads net gives additional emissions from traffic jams. Industrial production growth also (metallurgy, chemical, pharmaceutical, development, woodworking, radio engineering production, etc.)

*2.2. Model Description and Data Preparation*

In this research, the GRAMM/GRAL method (distributed with an open-source code under the GPL license) [34]. adjusted specially to concern the regional properties is used. The GRAL model allows simulating the pollutants particles dispersion from the sources along the airflow in the 3D model [35]. It means that the topography, surface properties, and buildings/construction are considered in the modeling. The primary airflows were calculated by the medium-scale meteorological model using the GRAMM algorithm based on the Reynolds-averaged Navier–Stokes equations (RANS equations) and the law of mass conservation [36]. Furthermore, the law of linear momentum conservation to calculate the movement of air volumes in the 3D modeling area's mesh cells is similar to CFD methods. It is important to mention that GRAL does not currently support chemical reactions and transformations between substances [37].

According to the state analytical report on the air quality in Krasnoyarsk [38], the period of AWC was 24 h from 6 December 2018 19:00 to 7 December 2018 19:00 (UTC +7). To take into account the background concentration, and for the model spin-up, the time interval of the simulation was 96 h (from 5 December 2018 00:00 to 8 December 2018 23:59). The hypothesis of the sufficiency of this interval was based on the following estimation: the maximum simulated distance in the experiment was 40 km—with an average wind speed of 2.3 m/s, the emissions from any source will cover it in about 5 h. The simulation domain properties are presented in Table 2.

**Table 2.** Simulation volume properties.

| Name | Value, Unit | Comments |
|---|---|---|
| GRAMM parameters | | |
| number of cells (l × w × h) | 454 × 418 × 20 | GRAM modeling area in cells |
| linear dimensions of the cell | 90 × 90 m | width and length of the cell |
| first layer height | 10 m | height of the first modeling layer |
| stretching factor | 1.16 | scale factor to underlain layer |
| height of the modeling area | 2506 m | height of the simulation area above sea level |
| GRAL parameters | | |
| number of cells (l × w × h) | 1118 × 779 × 80 | GRAL modeling area in cells |
| linear dimensions of the cell | 30 × 30 m | spatial resolution of modeling |
| first layer height | 2 m | at this level, the partial contributions of sources to pollution are fixed |
| stretching factor | 1.05 | scale factor to underlain layer |

Most of the parameters were selected based on the modeling domain size, the GRAMM/GRAL developers' recommendations (stretching factor and number of cells), and the computational capabilities.

Modeling in GRAMM/GRAL is performed at specified time intervals—1 h. The initial meteorological conditions such as wind speed and direction, and the atmospheric stability class are explicitly set for the beginning of each simulation hour (the initial values of meteorological parameters initialize for the entire simulation volume at the start of each hour). Then, the model for each cell calculates airflows based on the solution of the RANS equations, heat flows through the surface, and turbulence parameterization.

### 2.3. Improvement of the GRAMM/GRAL Model to Better Match the Regional Specifics

- Methodology upgrade.

Hourly recording of intermediate results (the pollutants concentrations from each group of sources) in every cell of the simulated volume and saving data on the atmosphere's state at different layers above the ground into the PostgreSQL database [39] were implemented.

- The vertical temperature gradient adjusting.

The coefficient of vertical temperature gradient was adopted through monthly-mean inversion parameters from Table 2.2.6 in [21]. The December mean inversion height is 900 m, and the temperature difference is 10.4 degree Celsius, so the inversion coefficient in the model was 11.56 degree/km.

- The river evaporation effect considering.

As mentioned earlier, the Yenisei River is not covered with ice during the winter seasons. A particular data file characterizing the entire modeling domain was prepared specially to take this phenomenon into account. Each sector represented in the CORINE [40] encoding based on the Land use data (describes below in Section 2.4.2.). The adjusted modeling algorithm controls each sector's indices, and in case of coincidence with the Yenisei index, the surface temperature is set to 4 Celsius degrees. Thus, the river evaporation effect in modeling was considered.

- Data preparation for the model verification.

Due to the relatively coarse 30-m grid, the cells with air quality measuring station (AQM) could be owned by an adjacent building, and it can distort the concentration values. In this case, the concentration of pollutants will be zero. It is also possible that the cell will be in the corner of two buildings, which will lead to irrelevant concentration values. GRAL's authors warn about the likelihood of such a problem in the "known issues" section of the model description [37]. To avoid such mistakes, a special cross-scheme was developed: if (x, y) are taken as the coordinates of the measuring point cell, then the averaging uses the values of the nearest cell with the following coordinates (x, y), (x−1, y), (x+1, y), (x, y−1) and (x, y+1); cells corresponding to buildings are excluded from the averaging. The algorithm smooths out the potential impact of irrelevance through averaging up to five cells; the diagram is shown in Figure 5.

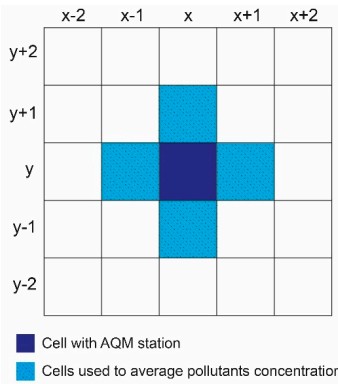

**Figure 5.** Averaging concentration values in a cell with AQM station for further model verification.

*2.4. Input Datasets and their Parameters*

The following sets of initial data were collected and preprocessed to initialize the model.

2.4.1. D Landform (Landscape) Model

The 3D model was based on the Enhanced Shuttle Land Elevation Data from Shuttle Radar Topography Mission (SRTM) [41] in GeoTIFF format [42] with 30 m spatial resolution. An adequate landscape layer with a spatial resolution of 3 m per pixel was obtained for further building placement through a bi-cubic spline interpolation in the QGIS platform [43]. The GeoTIFF was transformed to the ASCII (American standard code for information interchange) format [44], which was used in the model.

2.4.2. Land-Use Data and Landcover Vegetation Data

The landscape's physical properties such as surface reflectivity, surface emissivity, soil moisture, surface roughness, surface thermal conductivity, and surface heat transfer are fundamental for wind flow modeling. The Open Street Map (OSM) [45] data and classified Landsat-8 [46] and Sentinel 2 images [47] were used. An ASCII land-use layer with each sector in CORINE encoding concerning those properties was developed. The vector polygons characterizing the area covered by vegetation were obtained from this layer by "forest" tag in the "fclass" attribute. Then, each polygon was classified manually: for these reasons, a raster map of vegetation for the whole of Russia with a spatial resolution of 250 m [48] was used. The parameters such as vegetation height in m, trunk height in %, vegetation coverage in %, crown density in $m^2/m^3$, and trunk density in $m^2/m^3$ are considered in the simulation with default GRAL values.

2.4.3. Buildings

A special vector layer in Shp-format [49] characterizing the Krasnoyarsk civil and industrial development was designed based on the OSM datasets. For some buildings in the OSM, the height attribute was missing. That problem was solved by the information from the 2GIS system [50]: the building's height was calculated through the number of floors parameter and the coefficient 3.2 m per each floor (to consider the attics and technical spaces).

2.4.4. Meteorological Parameters

The weather conditions, which were used in this research to simulate meteorological fields, were obtained from the RIHMI-WDC [51,52] and represent the following hourly measurement information: wind speed and direction, surface, and air temperature. Data from the meteorological station with the international code 29572 (Krasnoyarsk downtown) on 5 December 2018 0:00 was used to set the initial simulation conditions. In particular, air and surface temperature −30 degrees Celsius, relative humidity—68%, ground temperature beyond a depth of 1 m −32 degrees Celsius. The main wind flow was also set according to this station's hourly data, whereas the air and surface temperature for every simulation hour were calculated by the model itself, based on surface properties and thermal flux balance (including solar radiance). Since the AWC meteorological conditions prevent pollutants dispersion in the city's atmosphere, the atmospheric stability class was accepted as 7 (the most stable). The wind speed characteristics for the entire modeling period are shown in Figure 6.

The graphs in Figure 6 confirm that the AWC mode ended at 7 December 2018 19:00 because the wind speed started to rise.

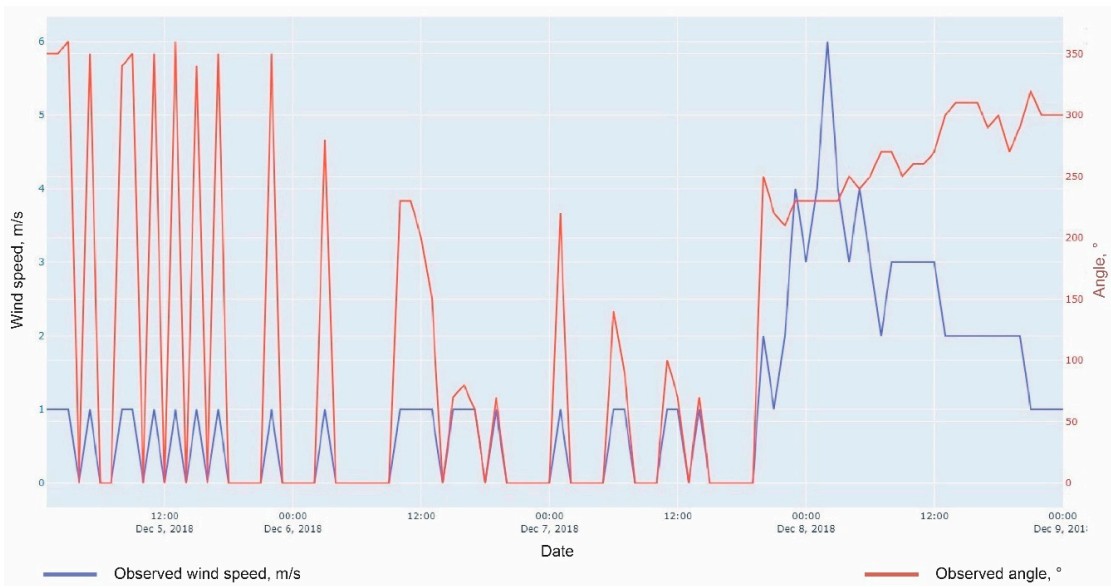

**Figure 6.** Wind speeds and directions by the data from meteorological station 29572 (Krasnoyarsk—downtown).

### 2.4.5. Air Pollutants

Modeling was carried out for the following emission elements: $NO_2$, $SO_2$, CO, Benzo(a)pyrene (BaP), PM10. According to the World Health Organization report [53], these substances make the most significant contribution to the air pollution in cities and agglomerations. In the Krasnoyarsk atmospheric surface layer, the excess of these substances is recorded quite often, according to the report of the Regional Ministry of ecology [17]. Modeling pollutants with main characteristics descriptions are presented in Table 3.

**Table 3.** The pollutants and their characteristics modeled in the experiment.

| Pollutant | Comments | Health Impact | Main Sources |
|---|---|---|---|
| Particulate matter (PM) | include both coarse (diameter <10 µm, PM10) and fine (diameter <2.5 µm, PM2.5) particles | pulmonary inflammation, acute nasopharyngitis, cardiovascular diseases, and infant mortality [54–56]. | coal-fired heating plants and private households, aluminum smelter, industry, cement production, building sector, motor roads |
| Nitrogen dioxide ($NO_2$) | as main and hazardous $NO_x$ component | respiratory diseases, coughing, wheezing, dyspnea, bronchospasm, and even pulmonary edema when inhaled at high levels [55] | fuel combustion, mostly heating plants and automotive engines exhaust |
| Sulfur dioxide ($SO_2$) | region-specific pollutant | rapid-onset bronchoconstriction, respiratory irritation, bronchitis, mucus production, and bronchospasm [56,57] | coal burning, industry |
| Carbon monoxide (CO) | has a significant part of gross emissions value | headache, dizziness, weakness, nausea, vomiting, and, loss of consciousness [54] | produced by fossil fuel when combustion is incomplete |
| Benzo[a]pyrene (BaP) | very hazardous region-specific pollutant | most carcinogenic polycyclic aromatic hydrocarbon (PAH) an important risk factor for lung cancer [54,55] | aluminum smelter, fossil fuel private households heating |

2.4.6. Sources of Pollutants

For the simulation, nine of the main groups of emission sources that most fully characterized the air pollution in Krasnoyarsk were identified: Central Heating Plants (CHP-1, CHP-2, CHP-3), Krasnoyarsk aluminum smelter and Krasnoyarsk metallurgical plant, local heating plants (boiler houses and steam shops), industrial enterprises, autonomous sources of heat supply within the city boundaries (the private households heated individually without a centralized system by lignite or firewood), transport/motor road network. The main groups of emissions sources are presented in Table 4.

**Table 4.** The main groups of emission sources.

| Group Index [1] | Group Name [1] | Description | Source Type |
|---|---|---|---|
| 1 | CHP—1 | Central Heating Plant—1, 485 MW, 1677 GCal/h | Point |
| 2 | CHP—2 | Central Heating Plant—2, 465 MW, 1405 GCal/h | Point |
| 3 | CHP—3 | Central Heating Plant—3, 208 MW, 582 GCal/h | Point |
| 4 | LHPs | Local Heating plants | Point |
| 5 | Industry | Enterprises—cement, asphalt, mechanical engineering, and metallurgical plants | Point |
| 6 | KAS | Krasnoyarsk Aluminum Smelter, the world second largest, about 1 Mt/y of aluminum | Point |
| 7 | Urban PH | Private households—Autonomous heating sources (AHS) that located in the Krasnoyarsk city borders | Area |
| 8 | Suburb PH | Private households—Autonomous heating sources (mostly villages and settlements) that located close to the Krasnoyarsk city borders | Area |
| 9 | Roads | Main transportation roads inside the Krasnoyarsk city | Linear |

[1] Index is used to identify the group in Table 5.

The information for Table 5 was obtained from [21]. This document contains both inventory and limits emissions values in g/s (gram per second) of all officially registered sources within the city boundaries (approximately at the end of 2016). Emission parameters were converted to kg/h, according to the GRAL requirements for input datasets, and aggregated in the groups listed in Table 4. Since [21] contains information on emissions from road transport based on a 2012 study, and emissions from private households are not included at all, additional calculations were performed to estimate actual emissions from these groups. Summary information on the emission parameters is presented in Table 5.

**Table 5.** The groups of sources and emission parameters in kg/h and % of gross total emissions.

| # [1] | $NO_2$, kg/h | $NO_2$, % GT [2] | PM10, kg/h | PM10, % GT | BaP, kg/h | BaP, % GT | $SO_2$, kg/h | $SO_2$, % GT | CO, kg/h | CO, % GT |
|---|---|---|---|---|---|---|---|---|---|---|
| 1 | 1061.8 | 21.9 | 2658.6 | 46.2 | 0.001360 | 0.44 | 2024.1 | 24.0 | 353.3 | 1.4 |
| 2 | 1328.8 | 27.5 | 278.9 | 4.8 | 0.000940 | 0.30 | 2222.6 | 26.4 | 115.9 | 0.4 |
| 3 | 485.6 | 10.0 | 82.8 | 1.4 | 0.000700 | 0.22 | 1382.4 | 16.4 | 68.6 | 0.3 |
| 4 | 624.2 | 12.9 | 956.0 | 16.6 | 0.001788 | 0.57 | 1203.4 | 14.3 | 2323.2 | 9.0 |
| 5 | 533.0 | 11.0 | 474.8 | 8.2 | 0.000395 | 0.13 | 355.4 | 4.2 | 1381.0 | 5.3 |
| 6 | 62.0 | 1.3 | 1017.4 | 17.7 | 0.264114 | 84.55 | 1142.5 | 13.5 | 7374.8 | 28.4 |
| 7 | 30.9 | 0.6 | 161.4 | 2.8 | 0.025329 | 8.11 | 56.5 | 0.7 | 7432.2 | 28.6 |
| 8 | 21.4 | 0.4 | 111.9 | 1.9 | 0.017560 | 5.62 | 39.2 | 0.5 | 5152.5 | 19.9 |
| 9 | 691.9 | 14.3 | 16.9 | 0.3 | 0.000207 | 0.07 | 8.3 | 0.1 | 1752.9 | 6.8 |
| GT [2] | 4839.5 | 100 | 5758.7 | 100 | 0.312392 | 100 | 8434.4 | 100 | 25954.4 | 100 |

[1] #—Group index is used to identify the group of sources in Tables 4 and 5. [2] GT—gross total.

### 2.4.7. Motor Road Network Source

The motor road network was modeled as linear emission sources. A vector layer in the Shp-format was prepared using the QGIS system based on the cartographic OSM and 2GIS data. Initial data of transport emissions were obtained from [21]. According to the official statistics [58], the number of vehicles in Krasnoyarsk in 2012 was 396,579, while in 2018, it was at least 411,000, so the emissions from each road network section was increased proportionally by a factor of 1.037. Every section of the road network was processed manually to estimate more accurately the contribution of this source to the surface layer air pollution.

### 2.4.8. Private Households Emissions

The territory of private houses, individually heated by stoves or boilers, covered about one-fifth of the entire Krasnoyarsk urban area. In the study, it was represented by a vector layer of polygonal objects, highlighting one-story houses (typical for Krasnoyarsk). For this layer verification, Earth remote sensing data (Landsat-8 and Sentinel-2 images), OSM, and Yandex maps service [59] were precisely processed by supervised classification. Special attention was paid to the 'residential' attribute to consider only the households of permanent people residence. The resulting household layer has an attribute—the number of buildings for each polygon. The structure and characteristics of emissions from the private sector are described in Table 5. Specific emissions of pollutants per unit of thermal source were calculated under Section 2 of the document [60]. The lignite fuel parameters (brown coal from Borodinsky open-cut mine [61]) and firewood (pine woods of 40% moisture) were used for calculation. The average private house area was used according to the data from local real estate agencies advertising to determine the required heating capacity; the energy needed to maintain a comfortable indoor temperature (+18 degrees Celsius according to local regulations) is taken into account in the calculation. The obtained results adjusted to the dimension kg/h for further use in modeling (Table 6).

**Table 6.** Emissions per one private household heated individually without a centralized system, kg/hour.

| # | Fuel Type | $NO_2$ | $SO_2$ | CO | BaP | Pm |
|---|-----------|--------|--------|--------|-----------|--------|
| 1 | Lignite | 0.0023 | 0.0068 | 0.2880 | 0.00000220 | 0.0125 |
| 2 | Firewood | 0.0014 | 0.0000 | 0.6120 | 0.00000086 | 0.0070 |
| 3 | 50/50 mix | 0.0019 | 0.0034 | 0.4500 | 0.00000153 | 0.0098 |

The average emissions value from both fuels was used in the simulation (50/50 mix) for every private house's polygon. This value was multiplied by the number of buildings in the polygon and normalized to its square.

### 2.4.9. Emission Sources Time Profiles

An essential requirement for the experiment was maintaining the maximum compatibility with the real emission parameters. For this reason, the time-intensity profile for each group of sources presented above (Table 4) was specially developed (Figure 7). The heating enterprises' intensity (CHP-1, CHP-2, CHP-3, boiler houses/steam shops) was constant and amounted to 100% of the maximum approved by the state regulations; the irregularity of emissions relative to the time of day from roads and private households was taken into account in the modeling under this profile.

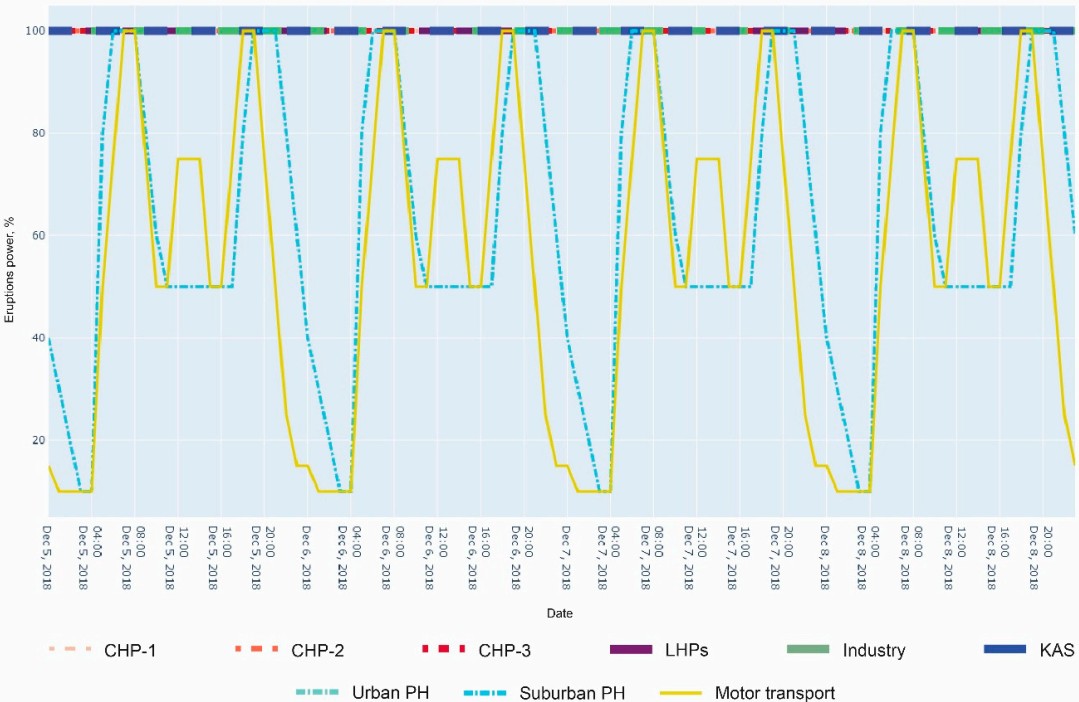

**Figure 7.** Emission sources time profiles designed to account for the intensity from different groups of sources.

### 2.4.10. Air Quality Observation Stations

Data from the regional air quality monitoring network was used to verify the simulation results [17]. Stations' coordinates were used as reference points to compare pollutants concentration values obtained through simulations with the instrumental measurement values.

## 3. Results

As the result of the experiment, for each cell of the simulated domain, the following datasets were obtained (raw-data after modeling):

- Speed and azimuth direction of wind;
- Partial contribution to the concentration of five pollutants from each of the main groups of sources listed above (Table 4).

These sets of values were fixed at the end for every hour for the entire simulation period—96 shapes of Krasnoyarsk atmosphere greater area stored. The pollutants hourly concentrations in cells corresponding to the atmosphere levels at 2, 10, 25, 50, 100, 150, 200, 250, and 500 m from the surface were obtained. Such a data structure allows building analytical layers, particularly vertical profiles of the atmosphere over the entire modeling domain (an example is given on Figure 8). In addition, it provides analysis of pollutant concentrations fluctuations in time—obtaining data at specific points for comparison with the measured values (at the AQM points).

The Supplementary Materials contain raw simulation results for constructing graphs and diagrams in a table format. Based on these data, several sets of summary materials were obtained (post-processing data); the Plotly Python graphing library (an open-source tool) was used for charts and diagrams plotting [62].

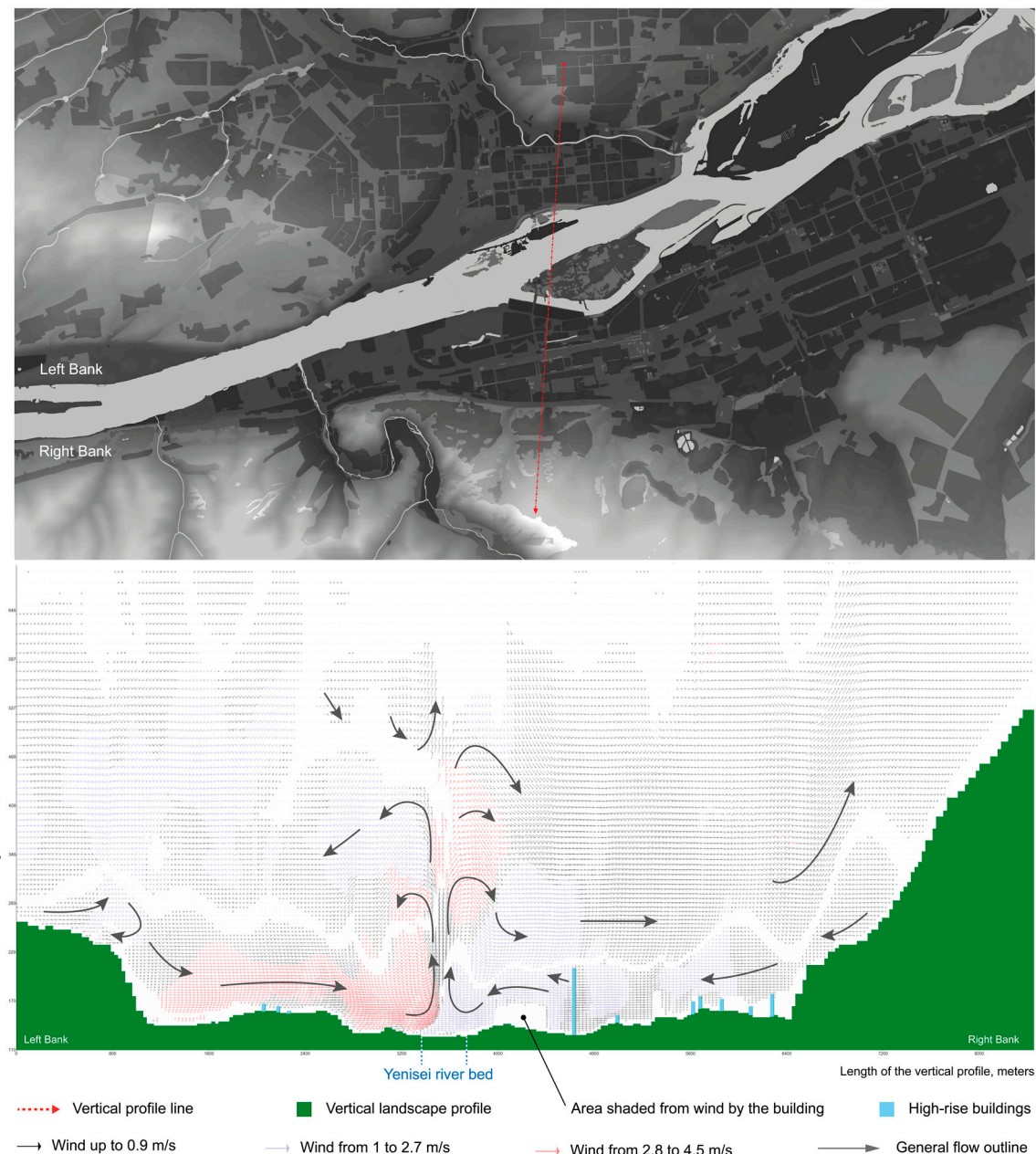

**Figure 8.** The Krasnoyarsk city atmosphere vertical profile is characterizing the Yenisei river beds influence on a calm air temperature of minus 25 degrees Celsius and water temperature plus 4 degrees Celsius.

## 3.1. Comparative Analysis of Wind Flow Trends—Simulated vs. Measured

The accuracy of pollutants dispersion modeling depends on the level of reliability of calculations of surface wind flows. [63]. In this context, the simulated and measured values were compared in the cell corresponding to meteorological station 29570 to evaluate the effectiveness of the dispersion model. This station is located in 11.9 km from the station used for initial meteorological simulation parameters (station code 29572, Figure 6). Cumulative wind rises for simulated (A) and measured (B) results are shown in Figure 9.

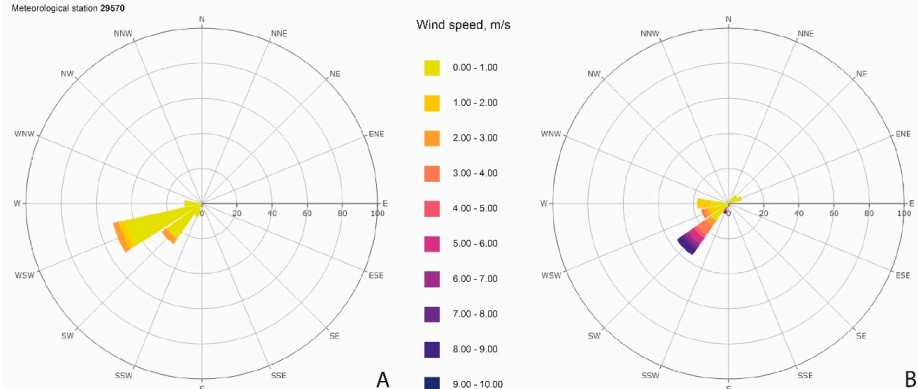

**Figure 9.** Simulated (**A**) and measured (**B**) wind flows and speeds distribution for meteorological station code 29570.

Charts with simulated and observed wind speeds and directions are presented in Figures 10 and 11.

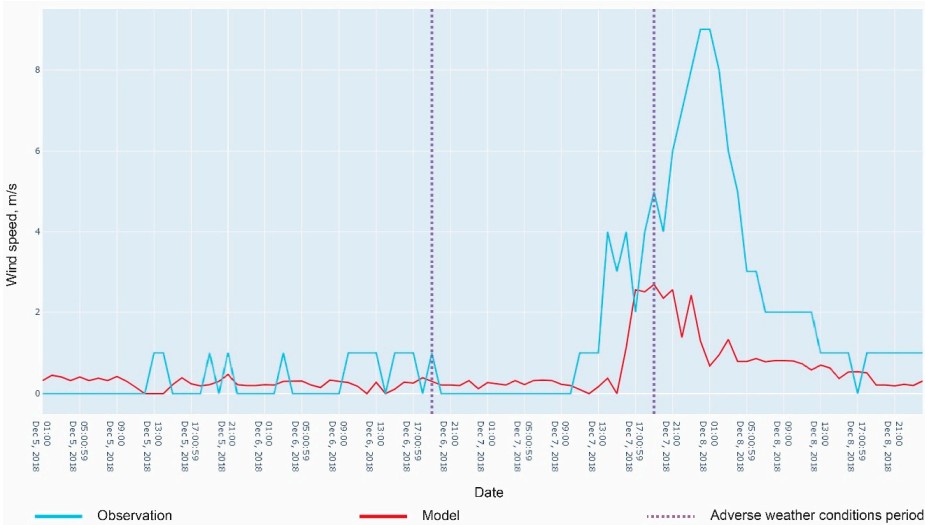

**Figure 10.** Simulated and observed wind speeds (meteorological station code 29570).

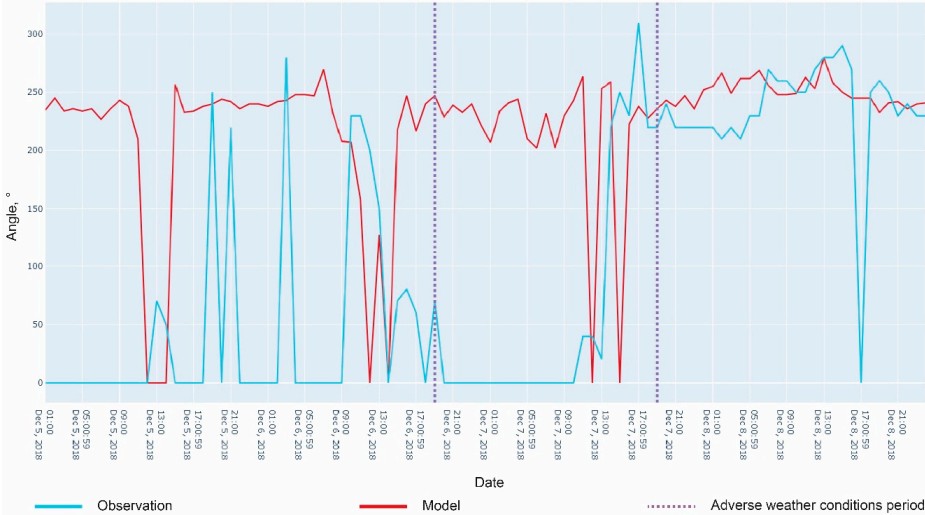

**Figure 11.** Simulated and observed wind directions (meteorological on code 29570).

The results of the experiment demonstrate that the model calculated even weak wind flows less than 0.5 m/s, while the minimum instrumentally determined speed was 1 m/s (blue line on Figure 10).

In addition, the direction of the flows with speeds less than 1 m/s was not instrumentally determined (blue line on Figure 11). The probable cause of these phenomena is the technical features of the station measuring equipment. Therefore, the low wind speeds before and during the AWC period impacted on the difference between the measured and simulated wind directions. In the second half of 7 December 2018, the wind flow increased and stabilized, and the wind speeds trends of simulated and instrumentally measured matched well. It could also be a remark that the model has underestimated the maximum wind speed. Such a behavior model was observed in the GRAL users workshop 2016 proceedings [64]. Notwithstanding the above, as a result, the total correlation for wind speeds is 65.3% and for directions, it is 68.6%.

### 3.2. Schematic Maps Characterizing the Emissions Dispersion for the Beginning and the End of AWC

The modeling results of BaP and PM 10 are presented in Figures 12 and 13; for other pollutants, please see the Supplementary Materials. These materials allow assessing the spatial distribution of air pollution emission plumes visually.

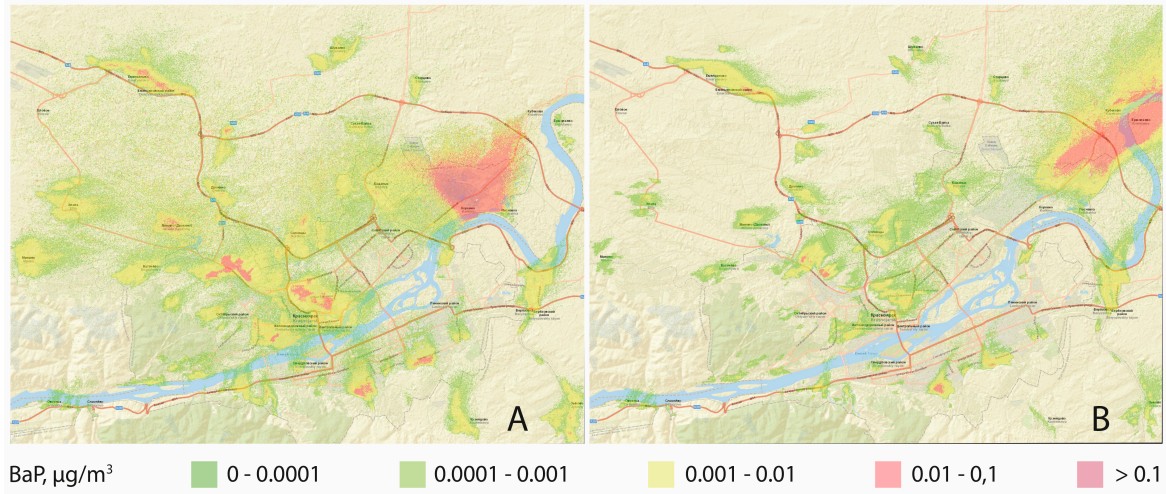

BaP, μg/m³ | 0 - 0.0001 | 0.0001 - 0.001 | 0.001 - 0.01 | 0.01 - 0,1 | > 0.1

**Figure 12.** Spatial distribution of benzo(a)pyrene (BaP) in the surface layer of the Krasnoyarsk atmosphere (2 m above the ground) at the beginning (**A**) and the end (**B**) of adverse weather conditions (AWC).

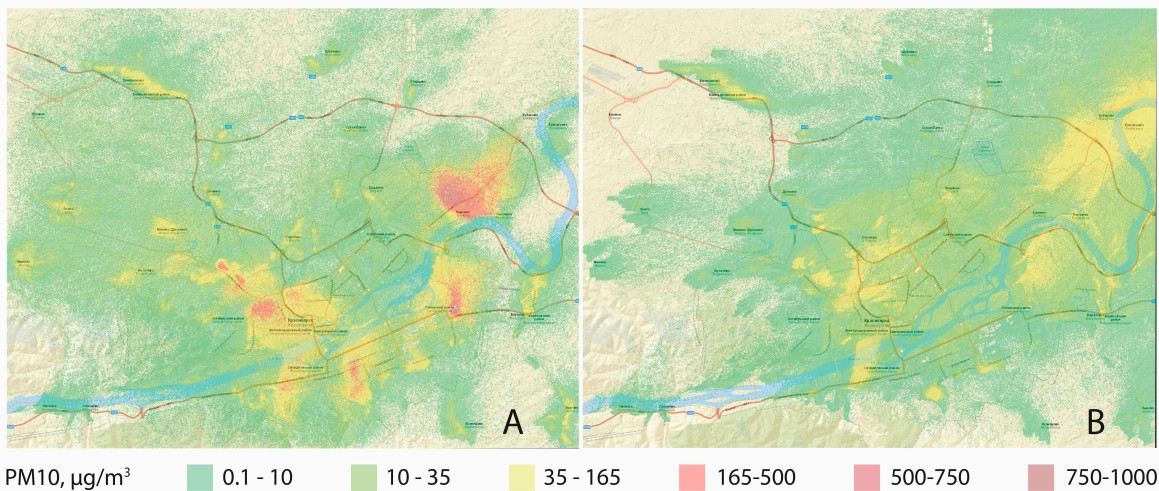

PM10, μg/m³ | 0.1 - 10 | 10 - 35 | 35 - 165 | 165-500 | 500-750 | 750-1000

**Figure 13.** Spatial distribution of PM10 in the surface layer of Krasnoyarsk atmosphere (2 m above the ground) at the beginning (**A**) and the end (**B**) of AWC.

### 3.3. Diagrams of Partial Contributions of the Main Groups Emission Source

The partial contributions of the main groups of emission sources to the surface layer pollution atmosphere are presented in Figures 14–16 (complete sets of schematic maps presented in Supplementary Materials). Pollutant concentration (in μg/m$^3$) for each group of source is plotted as a stacked area diagram. The orange background means that the total concentration exceeded the day mean MPC level, while the red background indicates that the total concentration exceeded the peak MPC level, and the green background indicates an acceptable level. Concentration contributions are presented in doughnut charts and separately marked for the beginning of the official AWC period (6 December 2018 19:00), the midterm of the AWC period (7 December 2018 07:00), and the end of the AWC period (7 December 2018 19:00).

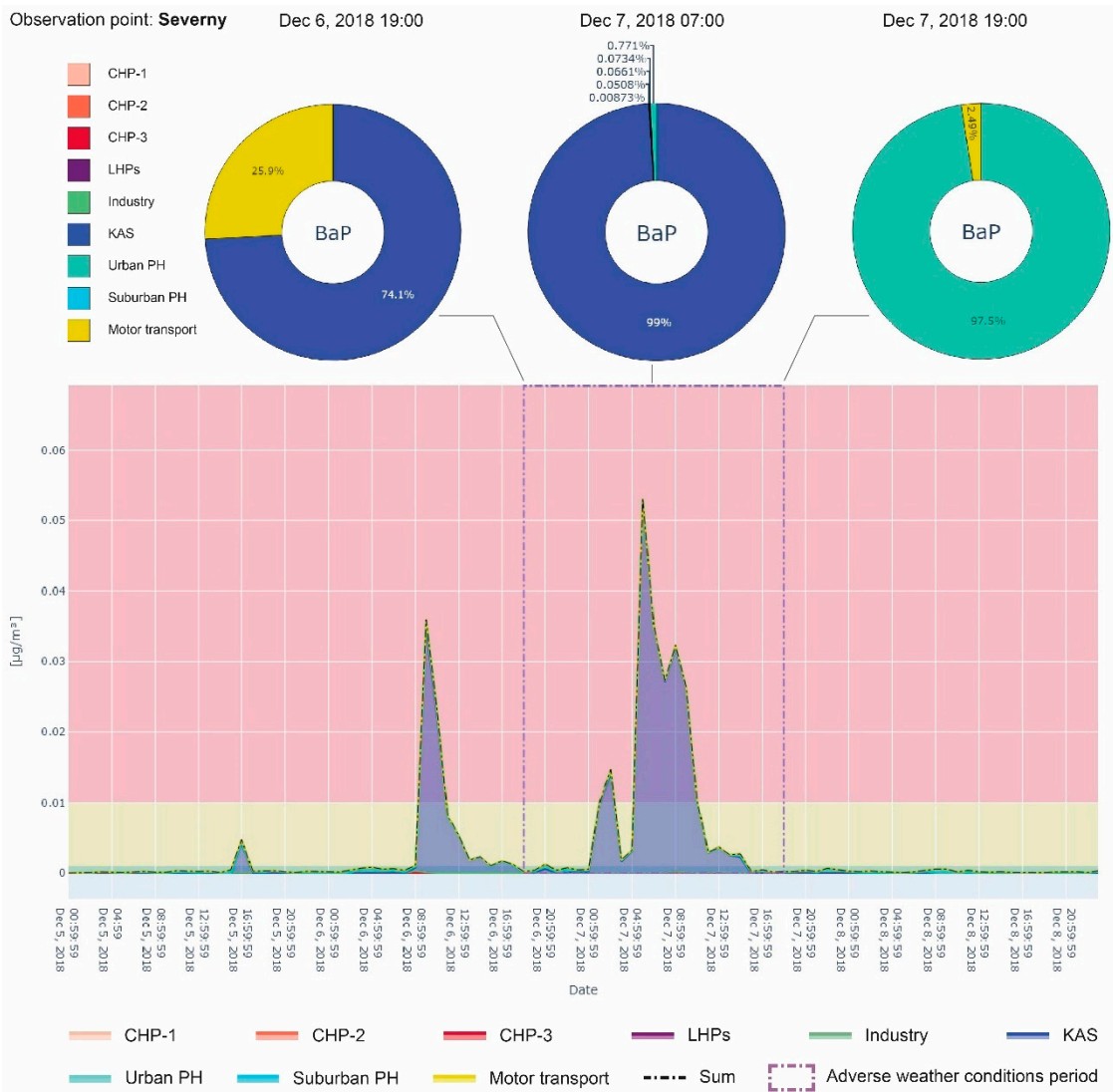

**Figure 14.** Simulated contributions of the main groups of emission sources in air pollution of BaP at 2 m height above the ground in the cells corresponding to the Severny observation point.

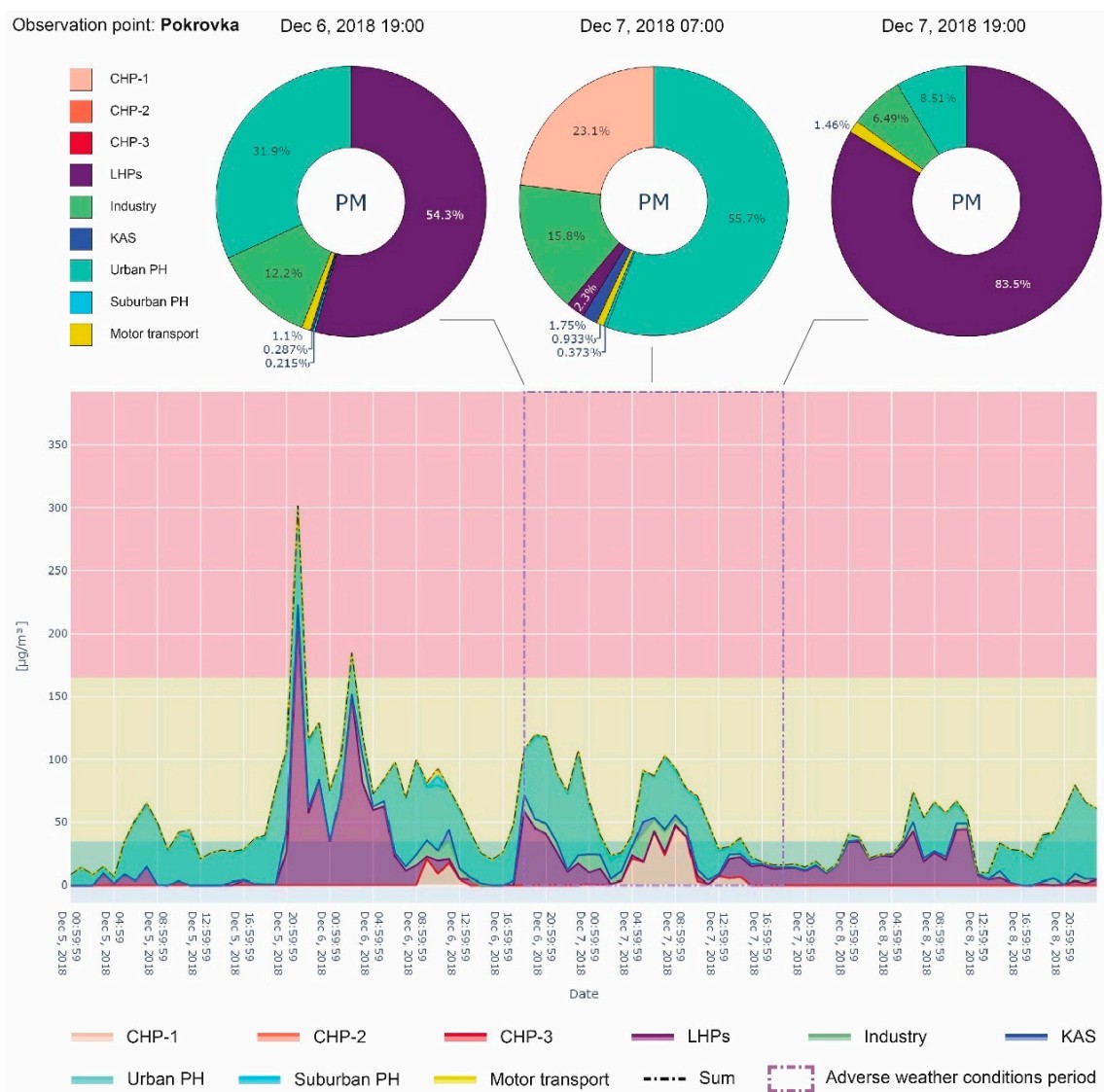

**Figure 15.** Simulated contributions of the main groups of emission sources in air pollution of PM at 2 m height above the ground in the cells corresponding to the Pokrovka observation point.

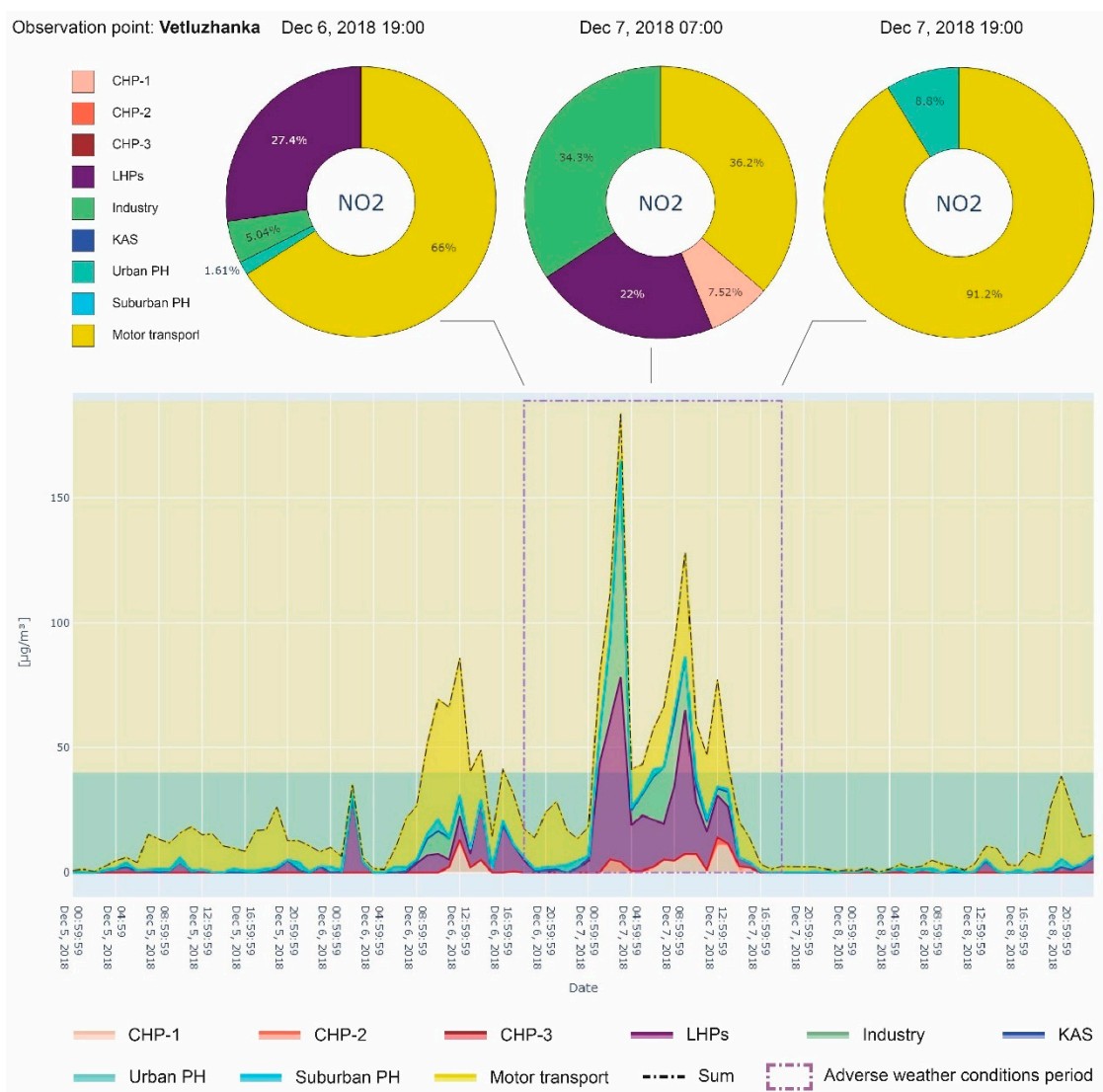

**Figure 16.** Simulated contributions of the main groups of emission sources in air pollution of NO$_2$ at 2 m height above the ground in the cells corresponding to the Vetluzhanka observation point.

*3.4. Comparative Tables for Each Pollutant Measured and Simulated Values*

Comparative tables characterized the quality of the air pollutant distribution modeling on 2 m above the surface were collected for each district of the city (Tables 7–10). Peak concentrations represent the maximum measured and simulated values at a specific time interval. The days on the eve and after AWC are defined by full 24-h periods (5 December 2018 and 8 December 2018); the AWC time interval is separated into the two parts (00:00–19:00 and 19:00–24:00 h).

**Table 7.** SO$_2$ concentrations in shares of peak maximum permissible concentration (MPC, 500 μg/m$^3$).

| City District | Date and Time Interval | | | | | | |
|---|---|---|---|---|---|---|---|
| | 05.12 00–24 | 06.12 00–19 | 06.12 19–24 | 07.12 00–19 | 07.12 19–24 | 08.12 00–24 | Result Mode |
| Cheryomushki | 0.03 | 0.03 | 0.03 | 0.03 | 0.03 | 0.05 | measured |
| | 0.38 | 0.18 | 0.04 | 0.04 | 0.04 | 0.11 | simulated |
| Severny | 0.13 | 0.16 | 0.14 | 0.14 | 0.08 | 0.10 | measured |
| | 0.02 | 0.36 | 0.01 | 0.37 | 0.03 | 0.02 | simulated |

**Table 7.** *Cont.*

| City District | Date and Time Interval | | | | | | |
| | 05.12 00–24 | 06.12 00–19 | 06.12 19–24 | 07.12 00–19 | 07.12 19–24 | 08.12 00–24 | Result Mode |
|---|---|---|---|---|---|---|---|
| Vetluzhanka | 0.01 | 0.01 | 0.01 | 0.01 | 0.01 | 0.01 | measured |
| | 0.02 | 0.20 | 0.01 | 0.22 | 0.00 | 0.02 | simulated |
| Solnechny | 0.09 | 0.64 | 0.74 | 0.99 | 0.00 | 0.02 | measured |
| | 0.01 | 0.28 | 0.10 | 0.30 | 0.01 | 0.00 | simulated |
| Pokrovka | 0.20 | 0.18 | 0.17 | 0.11 | 0.02 | 0.08 | measured |
| | 0.22 | 0.29 | 0.12 | 0.13 | 0.03 | 0.11 | simulated |

**Table 8.** CO concentrations in shares of peak MPC (5000 $\mu g/m^3$).

| City District | Date and Time Interval | | | | | | |
| | 05.12 00–24 | 06.12 00–19 | 06.12 19–24 | 07.12 00–19 | 07.12 19–24 | 08.12 00–24 | Result Mode |
|---|---|---|---|---|---|---|---|
| Cheryomushki | 0.50 | 0.66 | 0.58 | 0.76 | 1.06 | 0.10 | measured |
| | 0.17 | 0.16 | 0.04 | 0.10 | 0.06 | 0.23 | simulated |
| Severny | 0.74 | 0.94 | 0.84 | 0.96 | 1.18 | 0.16 | measured |
| | 0.12 | 0.30 | 0.11 | 0.27 | 0.08 | 0.10 | simulated |
| Vetluzhanka | 0.48 | 0.80 | 1.06 | 1.18 | 0.78 | 0.60 | measured |
| | 0.11 | 0.23 | 0.11 | 0.19 | 0.01 | 0.10 | simulated |
| Solnechny | 0.36 | 0.78 | 0.86 | 1.22 | 0.04 | 0.08 | measured |
| | 0.05 | 0.14 | 0.06 | 0.42 | 0.02 | 0.05 | simulated |
| Pokrovka | 0.74 | 0.68 | 0.70 | 0.44 | 0.00 | 0.22 | measured |
| | 0.68 | 0.74 | 0.66 | 0.62 | 0.04 | 0.66 | simulated |

**Table 9.** NO$_2$ concentrations in shares of peak MPC (200 $\mu g/m^3$).

| City District | Date and Time Interval | | | | | | |
| | 05.12 00–24 | 06.12 00–19 | 06.12 19–24 | 07.12 00–19 | 07.12 19–24 | 08.12 00–24 | Result Mode |
|---|---|---|---|---|---|---|---|
| Cheryomushki | 0.33 | 0.38 | 0.32 | 0.49 | 0.54 | 0.27 | measured |
| | 0.79 | 0.52 | 0.29 | 0.35 | 0.11 | 0.53 | simulated |
| Severny | 1.01 | 1.13 | 1.02 | 1.06 | 0.94 | 0.55 | measured |
| | 1.06 | 1.08 | 1.07 | 1.15 | 0.52 | 1.10 | simulated |
| Vetluzhanka | 0.01 | 0.05 | 0.05 | 0.03 | 0.02 | 0.04 | measured |
| | 0.13 | 0.43 | 0.14 | 0.92 | 0.01 | 0.19 | simulated |
| Solnechny | 0.12 | 0.00 | 0.00 | n/a [1] | 0.07 | 0.08 | measured |
| | 0.43 | 1.03 | 0.41 | 0.49 | 0.09 | 0.57 | simulated |
| Pokrovka | 0.48 | 0.34 | 0.37 | 0.37 | n/a | 6.49 | measured |
| | 0.75 | 0.75 | 0.38 | 0.45 | 0.10 | 0.30 | simulated |

[1] n/a—not available data.

**Table 10.** PM10 concentrations in shares of peak MPC (160 μg/m$^3$).

| City District | Date and Time Interval | | | | | | |
|---|---|---|---|---|---|---|---|
| | 05.12 00–24 | 06.12 00–19 | 06.12 19–24 | 07.12 00–19 | 07.12 19–24 | 08.12 00–24 | Result Mode |
| Cheryomushki | 0.03 | N/A [1] | N/A | N/A | N/A | N/A | measured |
| | 0.85 | 0.46 | 0.21 | 0.46 | 0.09 | 0.39 | simulated |
| Severny | N/A | N/A | N/A | N/A | N/A | N/A | measured |
| | 0.05 | 0.91 | 0.27 | 1.72 | 0.15 | 0.10 | simulated |
| Vetluzhanka | N/A | N/A | N/A | N/A | N/A | N/A | measured |
| | 0.09 | 0.91 | 0.06 | 1.53 | 0.01 | 0.12 | simulated |
| Solnechny | N/A | N/A | N/A | N/A | N/A | N/A | measured |
| | 0.02 | 1.11 | 0.08 | 1.30 | 0.03 | 0.03 | simulated |
| Pokrovka | 1.41 | 1.33 | 1.50 | 0.82 | 0.04 | 0.39 | measured |
| | 1.19 | 1.16 | 0.75 | 0.64 | 0.12 | 0.50 | simulated |

[1] N/A—not available data.

The tabular data format allows evaluating the quality of modeling for a specific pollutant in the spatial distribution context (concerning the city districts).

## 4. Discussion

The Graz Lagrangian model (GRAL) coupled with the Graz mesoscale meteorological model (GRAMM) calculated the energy and mass transfer in domain of 3852 km$^3$ distributed in 3,795,440 cells (according to Table 2) during 96 h. No adjustments to the intermediate results in the experiment that improve the accuracy of the model with the field measurements were made. Moreover, for each cell, the values of wind flows and concentrations were simulated at an acceptable level.

Pollutant dispersion mainly depends on air mass flows. GRAL's current research shows the model potential to correctly represent all majority processes in the lower atmosphere layer for pollutant dispersion prediction applicable to Krasnoyarsk Greater Area conditions.

Unfortunately, the state air quality report contains too little data for a matching analysis of simulated and observed values. An attempt to check the model to FAIRMODE (Forum for air quality modeling in Europe) [65,66] criteria compliance similar to the authors at [26] was made. However, due to the low number of observed values per monitoring station (from 1 to 6), the results are not relevant to the FAIRMODE methodology. For this reason, a specific assessment algorithm was developed:

- Satisfactory modeling criteria are achieved when the difference between simulated and observed concentrations values is below one fraction of the maximum permissible concentration pollutant.
- Outstanding model-to-observe fitness quality—when the amount of difference between the modeled and observed concentrations are below a one-tenth share of the maximum permitted concentration.

A comparative table of maximum (peak) and daily average concentration shows the numbers of matches of simulated and measured values for each pollutant (Table 11).

**Table 11.** The numbers of matches of simulated and measured values pollutants concentration.

| Pollutant | SO$_2$ | CO | NO$_2$ | BaP | PM10 | Total # |
|---|---|---|---|---|---|---|
| **peak MPC** | | | | | | |
| number of total mappings | 30 | 30 | 30 | 0 | 7 | 97 |
| number of satisfactory fitness (<1 shares of MPC) | 30 | 28 | 28 | 0 | 7 | 93 |

**Table 11.** *Cont.*

| Pollutant | SO$_2$ | CO | NO$_2$ | BaP | PM10 | Total # |
|---|---|---|---|---|---|---|
| number of outstanding fitness (<0.1 shares of MPC) | 18 | 7 | 10 | 0 | 1 | 36 |
| **day mean MPC** | | | | | | |
| number of total mappings | 10 | 10 | 8 | 14 | 2 | 44 |
| number of passably fitness (<1 shares of MPC) | 10 | 10 | 6 | 3 | 2 | 31 |
| number of outstanding fitness (<0.1 shares of MPC) | 2 | 3 | 0 | 0 | 0 | 5 |

In this research, a well-known two-factor assessment [64] was not applied, because the dataset has a significant share of observed concentrations in near-zero levels with relative values of the first tenth and hundredth shares of MPC. The normalization of the measured values was not applied either, since these conditions significantly impair the comparison of quality metrics. For example, matching simulated 0.03 and observed 0.01 values (in shares of MPC) gives a factor of threefold difference. Still, simultaneously, both of them are identical to the absence of noticeable emissions due to common sense.

The thesis about GRAL's suitability for use in the Krasnoyarsk conditions is confirmed by the experimental results presented above. Analysis of the results shows that the main contribution to the air pollution on the Krasnoyarsk surface layer is made by the local heating plants, autonomous heat supply, and vehicles. Whereas KAS and CHP-1 are leading in terms of gross emissions, their contribution to the atmosphere pollution is determined only in some areas (e.g., KAS especially for Severny and Solnechny city districts) and during AWC periods (according to this research case).

Any predictive model cannot predict the local movements of airflows in the near-surface layer with 100% accuracy, but the GRAL/GRAMM model showed a decent result for the Krasnoyarsk complex landscape and specific conditions. The following factors may have caused discrepancies between simulated and measured values:

- The initial meteorological conditions parameters were based on the data of only one observation point;
- Modeling spatial resolution was limited to 30 m;
- Emission sources descriptions and their modes of operation were based on the available official data (relevant for 2012–2017), part of the data was restored independently according to the period of 2019–2020; and emissions from unaccounted sources (unregistered small businesses) were not considered in the model (they can contribute significantly to pollution);
- Possible metrological problems at measuring points could affect the model verification.

The following points should also be clarified. Atmospheric conversion NO to NO$_2$ is not considered in the model that potentially underestimates simulated values. However, the inventory values of emissions for stationary sources in [21] already include NO to NO$_2$ conversion by following Appendix #5 in [11]. For the private sector, such a conversion was carried out by the calculation method [60]; for motor roads, it should be done by the [21] drafters as well.

Regarding AQM, the best convergence-simulated and measured pollutants concentration was obtained at the Pokrovka station, where the most considerable contribution was obtained from the private households. It can be argued that the specific data on emissions from private households (Tables 5 and 6) were calculated correctly (reflects the reality). In addition, GRAL accurately processes the spread of pollutants from low-pipe surface sources. Instrumental monitoring of NO$_2$ in Pokrovka demonstrates significant emissions on the last day of modeling (8 December 2018); however,

modeling did not confirm such a phenomenon. This discrepancy can be explained: the sensor did not work on the previous day, and start errors are not excluded.

The benzopyrene (BaP) observed average daily concentrations differ significantly across all stations with simulated results. However, in this regard, it is essential to note that the air sampling method for this pollutant significantly varies from all other substances (the results are obtained after the delivery and processing of air samples in laboratory conditions). According to the state environmental supervision requirements, the BaP average daily concentrations' determination requires a mandatory small sampling at the 01, 07, 13, and 19 h of the day, which falls on the emission peaks for variable sources (heating, traffic jams). In turn, this can lead to averaging over local maximum extreme concentrations, while the model averages overall hours in a day. Analysis of the simulated and measured values showed that the concentration discrepancies for these specific hours are less than the daily average. A possible source of increased concentration of benzopyrene at AQM posts, located near the residential sector, can be the exhaust gases of cars in a warming-up mode parked nearby. This regime is quite popular in Siberia during the winter period for the majority of the car park; the engines start automatically every 2–3 h to keep vehicles from freezing. In addition, the analysis of the measured concentration values showed slight overestimations for the CO and PM10. The balance of cold engine emissions in warm-up mode shifts from $NO_2$ to BaP [67], which is confirmed by the results of a comparative timeline analysis of the measured data at AQM posts. $SO_2$ is absent in significant quantities in emissions from vehicles. The best result of the convergence of simulated and measured concentration values was demonstrated for $SO_2$ and PM10 substances. The experiment showed that the sources of $SO_2$ and wind flows near them were calculated as efficiently as possible, since the initial data were worked out very precisely.

## 5. Conclusions

Methods for determining contributions to the surface layer air pollution, based on an estimate of gross emissions, are not suitable for areas with complex conditions (megacities, greater areas). They do not consider the nature of the pollutants dispersion and significantly distort the real causes of pollution. Particle tracking techniques are precise in assessing sources impact at a specific point, which guarantees more effective and adequate air quality control measures in cities. More accurate and objective approaches in the regulatory policy can encourage industrial companies to implement modern technologies for cleaning/reducing emissions and dust suppression on manufacturing. The GRAL model provides a sufficient level of reliability in assessing the pollutants concentrations in the surface layer and allows estimating the contributions of pollutant sources to the sum concentration at any point in the simulated domain.

The experiment needs to refine the initial data as well as a sufficient amount of instrumentally measured data for a comprehensive comparison and the model verification in accordance with the FAIRMODE criteria.

Directions for further work:

- Clarification of data on sources (especially for BaP), formation of a completer and a more up-to-date database;
- Use data from several weather stations throughout the modeling domain for a more accurate setting up of meteorological parameters;
- Modeling with a finer spatial mesh;
- Comparing the GRAL/GRAMM with Gaussian (and others) methods through a retro experiment;
- Development of methods for the verification of computational models based on remote sensing data.

**Supplementary Materials:** The supplementary materials are available online at http://www.mdpi.com/2073-4433/11/12/1375/s1.

**Author Contributions:** Conceptualization, A.A.R.; methodology, E.V.L. and A.A.R.; software, B.A.G., N.E.Z. and I.K.P.; validation, E.V.L., A.N.T. and A.A.R.; resources, A.S.V.; data curation, A.S.V.; writing—original draft preparation, A.A.R.; writing—review and editing, A.A.R. and E.V.L.; visualization, A.N.T. All authors have read and agreed to the published version of the manuscript.

**Funding:** This research received no external funding.

**Acknowledgments:** We thank the Technische Universität Graz, Dietmar Oettl and Markus Kuntner for developing and making open-access available the GRAMM/GRAL. We are grateful to the All-Russia Research Institute of Hydrometeorological Information and World Data Centre for the meteorological archives access. We thank the Ministry of Ecology of the Krasnoyarsk state for sharing emission information and pollutions observation data.

**Conflicts of Interest:** The authors declare no conflict of interest.

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
