# Peer review of "Graz Lagrangian Model (GRAL) for Pollutants Tracking and Estimating Sources Partial Contributions to Atmospheric Pollution in Highly Urbanized Areas"

_atmosphere, doi:10.3390/atmos11121375_

Round 1
Reviewer 1 Report
The manuscript depicts a critical environmental issue that is a complex source term estimation in Krasnoyarsk city's specific location (Russian Federation). The article is important and, in most aspects, scientifically solid. However, the introduction does not serve the entire manuscript properly. The authors succeeded in estimating the pollution levels in complex situations, but they do not provide a thorough comparison to results obtained using other models. Even a published detailed example in another city would help to support the idea made by the authors. A possible way to follow is to compare their results to the Gaussian model, which is being used by the local authorities. A proper introduction of the theoretical basis behind such a comparison in the introduction will make sense. The Gaussian model is very old and well documented; for this approach to work, the authorities' way of implementing it should be interesting for the experts in the research area. Since I'm not familiar with the specific version of the authorities' model, it is hard to tell if this approach makes sense. However, the idea behind this concern is that reporting a single case with a single set of parameters without addressing any other possibilities may be unfavorable. I have some other concerns listed below.
|
Line # |
Comment |
|
42, 75, 108, 110, 112 and more |
First body use (We) |
|
53 |
The PUFF model presentation should be presented after the Lagrangian model description. |
|
40-69 |
It is hard to tell what is the purpose of the various atmospheric dispersion model description. Repeating known information is redundant unless it supports a specific point made in the manuscript. For example, a new approach is presented that overcomes the shortcomings of state-of-the-art methods. Currently, the methods are presentation seems too short and pointless. This section should be revised – make it shorter or longer. It has to have an effect on the paper. |
|
70 |
1. Should mention that these methods utilize an atmospheric dispersion model. 2. The discussion is simplistic and misses some details (A forward-backward coupled source term estimation). Quote: https://doi.org/10.1016/j.anucene.2017.01.039
|
|
Introduction |
The introduction does not support the rest of the manuscript. The researchers used GRAL, which is a specific implementation of the Lagrangian. They better focus their introduction on this specific implementation's properties or compare it to the Gaussian dispersion model currently being used by the authorities or other models in the same or at least similar conditions. It is reasonable to assume that they do not use the Gaussian dispersion model in its simple form rather work around it (Aloha?). Such an introduction will better serve the manuscript. I found this sentence in section 2.1.: "The methods of assessing the impact on cities atmosphere quality used in Russia are not suitable for the conditions of the Krasnoyarsk due to specifics described above. As a result, management decisions are often based on estimates of gross emissions that do not reflect the reality and ultimately, they are ineffective." It may help to have it with more detail in the introductory part. |
|
188 |
I didn't understand: "We adjusted they specifically for the regional properties of the above as the basis of the algorithm development for numerical experiment." |
|
202 |
Table 2, why did you choose these parameters? Did you try other sets of parameters? |
|
358-376 |
Meteorological Parameters: I understand that: "The main wind flow was also set according to the data of this station hourly, whereas air and surface temperature for every hour of simulation were calculated by model itself, based on surface properties and thermal flux balance (include solar radiance)." So, Graphs 9, 10, and 11 compare meteorological conditions at different locations. Am I right? If so, the (large) deviation between calculation and observation is acceptable and should be addressed more clearly. |
Reviewer 2 Report
The manuscript presents an interesting and extended research to apply an air quality assessment model in a severly polluted city. This impressive work deserves a better presentation, both in terms of language and the description of methods. Extensive English language editing is required. A non-exhaustive list of proposed grammar improvement is given at the end of the review.
Specific comments:
Modeled and measured meteorological values are simultanously referred to in the manuscript. How are the measured meteorological parameters assimilated in the model?
line 339: Why were wind speed and velocity set constant? This contradicts to the results presented in Figs. 10-11.
Please provide more detail on the derivation of parameters in Table 5. A few methods are given for households and traffic, but there is much more in Table 5 to be discussed as these are critical input values for the model.
“a mature meteorological CFD system” – no CFD capabilities were mentioned throughout the manuscript until this point. According to the 30 m mesh, it indeed looks like a CFD model, but please provide much more detail on the CFD model setup (especially boundary conditions and turbulence modeling).
„the modelling did not consider the background concentration values” – did you consider applying a spin-up period of the model to construct the initial air pollution field? While it still does not solve the large-scale background issue, at least it creates a local background from the previous few days emissions. In stagnant events like the AWC investigated here, a few spin-up days should provide a reasonable background pollution.
line 478: a further limitation is the negligence of secondary particle formation. A few remarks should be added on this issue (why and how much is the secondary part of PM pollution negligible?).
line 529: I’m not sure a mesh finer than 30 m would really improve the results. Instead, I would look more into different turbulence schemes and parameterizations.
Comments on grammar:
Please provide units in Table 11.
line 11: studying Air quality
line 19: km
line 40: used to -> is applied to
line 165: global Air quality
line 176: please review phrasing
line 188: adjusted them
line 248: please use superscript
line 257: simulation
line 262: ground temperature
line 272: because
line 276: World Health Organization
line 280: are presented
line 347: analysis of fluctuations
line 362: is located
„Reference source not found!” in line 387
line 399: are visualized, is presented
line 400: please use superscript in unit
Reviewer 3 Report
General comments
The paper “Graz Lagrangian Model (GRAL) for pollutants 2 tracking and estimating sources partial contributions 3 to atmospheric pollution in highly urbanized areas.” presents the setup, application and validation of the GRAMM/GRAL modelling system to evaluate air quality and perform source apportionment analysis in the Krasnoyarsk area (Russia)
The content of the paper is in line with the scope of Atmosphere and refers to a relevant issue of air quality modelling. Authors performed an interesting and thorough modelling analysis concerning an heavily polluted area.
However, at this stage, the paper is very far from to be suitable for publication for several mains reasons that authors should carefully consider in a revised version:
- The introduction need to be deeply reviewed with respect to contents and syntax. The first part of the introduction provide a generic and not always appropriate overview of the main modelling approaches; but all these concepts are well known and not particularly related to the goal of the paper. In my opinion, authors should reorganize the introduction in order to describe:
- The main features of the problem that are going to study (region, main pollution problems, previous studies,…)
- the modelling approach they want to apply, according to the main issues they want to tackle
- the main goals of their work
- the elements of novelty, also in terms of possible differences/improvements with respect to previous modelling studies
- the organization of the paper
- English language is very poor, mostly unclear in the use of sentences and wording and often presenting syntax errors. A full review by mother tongue people is suggested
- The presentation of the key features of the modelling system and the modelling setup is rather poor and confusing and it is very hard to understand the key assumptions implemented by authors
- There are some errors in formulas
- Model performance evaluation and corresponding discussions are uncomplete (see details below)
For these reasons the paper should be rejected as it will require a deep revision before being considered again for submission.
In the following further comments and suggestions are detailed
Specific comments
In the following just a few comments on a possible reorganization of the paper contents.
R19 – “KM3” ?
R29 – “model performance evaluation” instead of “match to observed”?
R34 – “on the pollution within the lowest atmospheric layers”?
R34-37 – rephrase the statement
R40 – The verb is missing in the statement
R41 – “the speed of obtaining the result” -> to be rephrased
R42 – “variability” instead of “variance”?
R50 – “Gaussian models are based”
R53 – “Lagrangian” not introduced yet
R57-59 – The definition of lagrangian model is not adequate. It is seems more related to trajectory models that, likewise PUFF models belongs to the more general family of “lagrangian models”
R70-73 – Not clear what authors would like to point out with this statement
R119 – “…weather conditions (AWC), characterized by weak….”
R120 – verb is missing
R130 – verb is missing..
R145 – formula 1 is unclear: authors defines Vm, but formula includes Vmst’, what is “j”? the month number? If yes, should Kj be defined as Kdj -> number of non zero measurements for day “d” in month “j”?
Likewise what does Nj represent? Should it represent the number of days in month “j”? etc…
R150 – if definition for Kj and Nj are correct for formula 2, it seems they do not hold for formula 1
R165-172 – The underlying concepts are clear but syntax should be strongly improved.
R185 – “observed data”
R190 – “dispersion” instead of “distribution”?
R192- The description of the GRAMM setup is rather confusing: what are “primay airflows”? In which sense they are similar to CFD methods?
R201 – “domain” instead of “volume”
R214-221 – Which aspect of the modelling chain is influenced by the corrections authors implemented in land use data? Vertical dispersion coefficients? Others?
R222-228 – Authors implemented a rather unusual interpolation method to compare modelled and observed data. The most common approaches are:
Taking into account only the concentration of the cell that includes the AQM station (the blue one in figure 5), generally known as “nearest cell”
Considering the 4 cells whose centres enclose the AQM site and the performing a bilinear interpolation of the 4 modelled concentrations
Why do authors apply the approach based on figure 5?
R257 – “The weather conditions used TO SIMULATE METEOROLOGICAL FIELDS were obtained…”
R270 – “Wind speed and DIRECTION…”
R275 – Did authors consider all pollutants as non-reactive or did they consider chemical transformations? Additional details are needed.
R281 – Table3 – Does Road transport not represent a relevant PM source?
R289 – “eruptions”??
R290 – it seems that authors refer to a document including emission “limits” more than an emission “inventory”. Any comment on this issue?
R305 – what do authors mean with “one-story objects”?
R362 – Which station?
R376 – “wind direction”
R382 – any additional comment on the strong wind speed underestimation taking place from mid morning of December 7th?
R387 - error in Word crossed reference… Figures number is partially lost.
R426 – why not using the absolute concentrations instead of a share of pollutant peak?
R435-437 – what’s the meaning of this statement?
R445-451 - Authors tried to implement a different set of metrics to evaluate their model performance. This represents an interesting exercise but the explanation is rather confusing. Moreover the use of more traditional indicators such as BIAS, RMSE, correlations, etc… would help in understanding model results
R452 – Table 11 requires additional comments and details.
Round 2
Reviewer 1 Report
Nicely revised, good luck with your work
Reviewer 3 Report
Authors carried a thorough and careful review, substantially increasing the quality of the paper. They properly addressed all comments and requests of correction and clarification.
They also significantly improved the English language, making the paper easily readable and effective in presenting the main results.
In my opinion, there are are just a few very minor issues that could be solved before final publication, namely:
R131 – There are still a few minor bugs in formula 1, namely:
- Vmst is not explicitly defined
- Rj is the number of wind speed measurement every 3 three hours from 1966 to 2019, FOR MONTH J?
- “s” and “qj”… what exactly do they account for?
R282-285 – It seems there is a repetition with R272-275
R477-478 – This assumption could be very critical, because –generally- NO account for the major fraction of NOX emissions (around 95%) and then it is converted to NO2, through rather fast reactions. Therefore , how did authors handle this issue? Did they assume that NOX emissions were considered as NO2? Other?
